# Transcriptomic landscape of the blastema niche in regenerating adult axolotl limbs at single-cell resolution

Nicholas D. Leigh[1,2], Garrett S. Dunlap [1,2], Kimberly Johnson [1,2], Rachelle Mariano[1], Rachel Oshiro[1], Alan Y. Wong [1,3], Donald M. Bryant[1], Bess M. Miller[1,2], Alex Ratner[4], Andy Chen[1], William W. Ye[1], Brian J. Haas[2] & Jessica L. Whited[1,2,3]

Regeneration of complex multi-tissue structures, such as limbs, requires the coordinated effort of multiple cell types. In axolotl limb regeneration, the wound epidermis and blastema have been extensively studied via histology, grafting, and bulk-tissue RNA-sequencing. However, defining the contributions of these tissues is hindered due to limited information regarding the molecular identity of the cell types in regenerating limbs. Here we report unbiased single-cell RNA-sequencing on over 25,000 cells from axolotl limbs and identify a plethora of cellular diversity within epidermal, mesenchymal, and hematopoietic lineages in homeostatic and regenerating limbs. We identify regeneration-induced genes, develop putative trajectories for blastema cell differentiation, and propose the molecular identity of fibroblast-like blastema progenitor cells. This work will enable application of molecular techniques to assess the contribution of these populations to limb regeneration. Overall, these data allow for establishment of a putative framework for adult axolotl limb regeneration.

[1] Department of Orthopedic Surgery, Harvard Medical School, The Harvard Stem Cell Institute, Brigham and Women's Hospital, 60 Fenwood Road, Boston, MA 02115, USA. [2] Broad Institute of MIT and Harvard, 7 Cambridge Center, Cambridge, MA 02142, USA. [3] Department of Stem Cell and Regenerative Biology, Harvard University, 7 Divinity Avenue, Cambridge, MA 02138, USA. [4] ICCB-L Single Cell Core, Harvard Medical School, 200 Longwood Avenue, Boston, MA 02115, USA. Correspondence and requests for materials should be addressed to J.L.W. (email: jessica_whited@harvard.edu)

Many salamanders, such as axolotls, have the remarkable capacity to regenerate entire multi-tissue structures, such as limbs, throughout their lives. This is in stark contrast to mammals, which have extremely limited capacity to regenerate multi-tissue structures. After amputation of an axolotl limb, a clotting response occurs, and the wound is quickly covered by the migration of a specialized wound epidermis (WE)[1]. The WE can be broken down morphologically into an outer layer of apical cells, a thicker intermediate WE, and a columnar basal layer[2]. Underneath the WE, progenitor cells aggregate and form what is called the blastema. The blastema is a combination of lineage-restricted and multipotent progenitors that gives rise to the internal structures of the regenerated limb[3–6].

The interaction between the WE and blastema is integral, and a variety of techniques have shown that the WE is required for limb regeneration[7–9]. This requirement is dependent on roles in promoting blastema cell proliferation[10], stump tissue histolysis[11], and guiding blastema outgrowth[12]. In addition to contributions from the WE, macrophages and nerves are required for limb regeneration[13,14], highlighting that a coordinated effort between multiple cell types is required for blastema formation. Blastema is a broad label for the collective organization of possibly de-differentiated dermal fibroblasts and periosteal cells, Pax7+ muscle satellite cells, and hitherto undiscovered populations that contribute to limb regeneration[4–6,15,16]. A deeper understanding of the cell populations present in regenerating limbs, especially during the early stages, is important for understanding the activation, recruitment, and differentiation required to create blastema cells. Previous studies have been instrumental in providing information about gene expression across the course of limb regeneration (reviewed in ref. [17]). However, these studies used bulk RNA-sequencing (RNA-seq) approaches, yielding composite measurements, and therefore identification of pivotal cell type-specific transcripts with unique gene expression could be masked.

Recently, with the advent of single-cell RNA-seq an unexpected diversity of cellular subtypes has been uncovered even within well-delineated systems[18–20]. Most work on single-cell RNA-seq has been dedicated to systems with a wealth of pre-existing knowledge about the cellular composition, aiding in the description of previously described and undescribed cell types. In contrast, there is a limited understanding of the diversity of cells and their behaviors during axolotl limb regeneration. Thus, we undertook an unbiased and comprehensive analysis of the cell populations that contribute to axolotl limb regeneration by performing single-cell RNA-seq on over 25,000 cells from the limb at homeostasis and at multiple time points during limb regeneration (Supplementary Table 1). Here, we focus on the early stages that are required to build and grow a blastema. We propose differentiation trajectories for both epidermal and mesenchymal cells during regeneration. Our study provides an important resource that defines the molecular identities of the cell populations present in the regenerating limb, opening the way for future study of these cell populations and their roles during axolotl limb regeneration.

## Results

### Cellular heterogeneity of the regenerating limb.
The regenerating limb contains different cell types that can be distinguished morphologically, including: blastema cells; the basal, intermediate, and apical layers of the WE; and erythrocytes (Fig. 1a, b). To evaluate the cellular composition of the regenerating limb at the molecular level, we performed single-cell RNA-seq over a time course of adult axolotl limb regeneration. We sampled tissue from homeostatic limbs (to establish a ground state) as well as three regenerative milestones—wound healing, early-bud blastema, and medium-bud blastema stages. The early-bud blastema constitutes a time point when progenitor recruitment and activation are occurring, and the medium-bud blastema is after activation and mobilization of progenitor cells from the stump, but prior to any outward re-differentiation into the regenerated limb. We collected the entire regenerate, including both the WE and blastema, to provide comprehensive snapshots of the cell populations present at each regenerative stage with two to six biological replicates per time point. We utilized inDrops, a high-throughput microfluidic platform, to capture and subsequently sequence thousands of single cells[21,22] (schematic in Fig. 1c).

We sequenced over 25,000 cells from 13 separate homeostatic and regenerating forelimb samples, derived from 13 adult axolotls. Using the Seurat single-cell analysis toolkit[23], we performed dimensionality reduction analysis, unbiased clustering of cell populations according to similar gene expression patterns (i.e. no prior knowledge of cell population markers is required to drive clustering), and visualization of resulting clusters in two dimensions via t-distributed stochastic neighbor embedding (t-SNE). We recovered between 11 and 19 transcriptionally distinct populations depending on time point sampled. Based on transcriptional profiles, we hypothesize that these cell clusters consist of multiple populations of fibroblast-like blastema (FLB) cells, tenocytes, Pax7+ muscle satellite cell-derived myogenic blastema cells, endothelial cells, erythrocytes, myeloid cells, lymphoid cells, and an epidermis composed of a variety of distinct populations (Fig. 1d, Supplementary Data 1–5). Importantly, nearly all these populations were present at homeostasis and during regeneration, including a population of fibroblasts that shared similarity to putative FLB cells at all regenerative time points sampled (Supplementary Data 1–5). This transcriptional profiling at single-cell resolution allowed for identification of conserved marker genes that define cell populations, elucidation of phenotypic changes in these cell populations that are associated with regeneration, and exploration of differentiation trajectories.

### The axolotl epidermis is heterogeneous.
Initiation of blastema formation is dependent on the formation of a WE that is regeneration competent. Of interest are differences in the cellular composition and changes in gene expression in conserved cell populations between regenerating versus homeostatic epidermis. At homeostasis, we found unique populations that we postulate to be ionocytes expressing *forkhead box i1* (*foxi1*) and *atpase h + transporting v1 subunit b1* (*atp6v1b1*)[24] (Supplementary Fig. 1a, Supplementary Data 2) and putative epidermal Langerhans cells defined by high expression of histocompatibility genes, such as *major histocompatibility complex, class II, dr beta 1* (*hla-drb1*), with co-expression of epidermal genes (i.e. *keratin 12* (*krt12*))[25] (Supplementary Fig. 1b, Supplementary Data 2). In homeostatic epidermis, we uncovered four other populations of epidermis, which based on marker gene expression we postulate to be the basal epidermis (*collagen type xvii alpha 1 chain* (*col17a1*))[26], proliferating epidermal cells (high levels of *proliferating cell nuclear antigen* (*pcna*)) (Supplementary Fig. 1c), intermediate epidermis (*krt12*)[25] (Supplementary Fig. 1b, bottom), and small secretory cells (SSCs, *otogelin* (*otog*) and *fc fragment of igg binding protein* (*fcgbp*))[27] (Supplementary Fig. 1d, Supplementary Data 2). Our data indicate an absence of both ionocytes and putative Langerhans cells in the WE, while SSCs, basal epidermis, and populations of intermediate WE are present at all regenerating time points (Supplementary Data 1). Notably, during wound healing, we identified a cell population we postulate to be Leydig cells based on expression of *chitin synthase 1* (*chs1*)[28] and *anterior gradient protein 2-a* (*agr2a*)[29] (closest match to newt

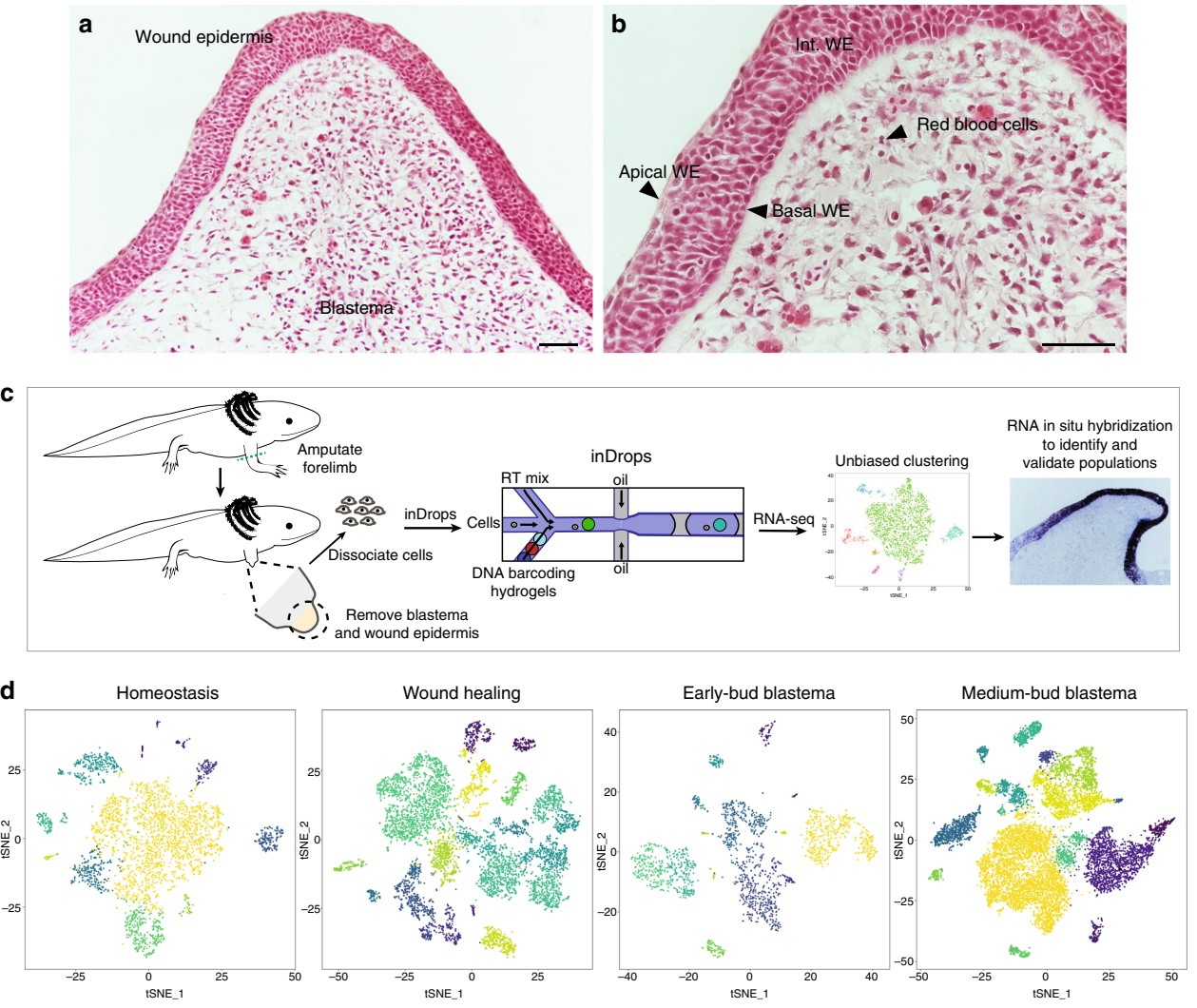

**Fig. 1** Single-cell RNA-seq of the adult axolotl limb. **a** Hematoxylin and eosin staining of medium-bud blastema with overlying wound epidermis. Cell types labeled in **a**, **b** based on morphology/location. Scale bars are 100 μm, $n = 4$ animals. **c** Strategy for identifying the cell populations that compose the regenerating axolotl limb. **d** t-SNE plot of cells collected from homeostasis ($n = 2$ animals), wound healing ($n = 3$ animals), early-bud blastema ($n = 2$ animals), and medium-bud blastema ($n = 6$ animals). Each dot is a cell and colors represent individual populations and do not reflect the same population between time points

AGP) that was not obviously present other time points (Supplementary Fig. 1e). At regenerating time points, we find multiple populations of what we deem intermediate WE, due to a lack of unique transcripts (Supplementary Data 3–5). These populations likely represent cells along a differentiation trajectory, which we explore further below.

We observe regeneration-specific gene expression in populations such as the basal WE, which has long been thought to play an essential role in crosstalk between the blastema and WE[9] (Supplementary Fig. 1f). We also evaluated the expression of previously described factors that are expressed in, or near, the WE and have been reported to influence regeneration, specifically AGP[29], dlx3[30], and nrg1[31]. In accordance with the literature, we observe expression profiles similar to the published reports (Supplementary Fig. 2a–c). Importantly, we observe very few nrg1 transcripts in our study, which is likely due to our protocol that would not allow for capture of nerve cells, as only axons project into the limb, not the cell body.

We next sought to confirm the existence of cell population-specific and regeneration-specific genes as identified by the single-cell RNA-seq dataset (Fig. 2a, e). We performed RNA in situ hybridizations on krt12, a pan-epidermal gene (Fig. 2b, f); otog, a gene enriched in SSCs, a specific epidermal population (Fig. 2c, g); and frem2, a gene with both regeneration- and time point-specific expression (Fig. 2d, h). These RNA in situ hybridizations confirm the expression profiles generated from the single-cell RNA-seq data (Fig. 2a, e).

To begin to understand how the WE is formed, we pursued the idea that the axolotl epidermis follows an outward differentiation trajectory as seen in other species[32]. If axolotl epidermis recapitulates the differentiation trajectory of mammals, we would expect to detect an epidermal stem cell reservoir feeding into a differentiation trajectory culminating in cells residing in the outermost layer of the epidermis. This could be the case for both homeostatic epidermis and the WE, and thus we tested these two ideas separately. To do this, we performed pseudotime analysis using Monocle[33]. Construction of a pseudotime trajectory allows for ordering of cells across a learned trajectory (i.e. determining where a cell is in the process of differentiation). For both sets of analyses, we included populations of basal and intermediate epidermis, as well as SSCs. We chose to omit rare populations (e.g. ionocytes with 27 total cells) and proliferating epidermal

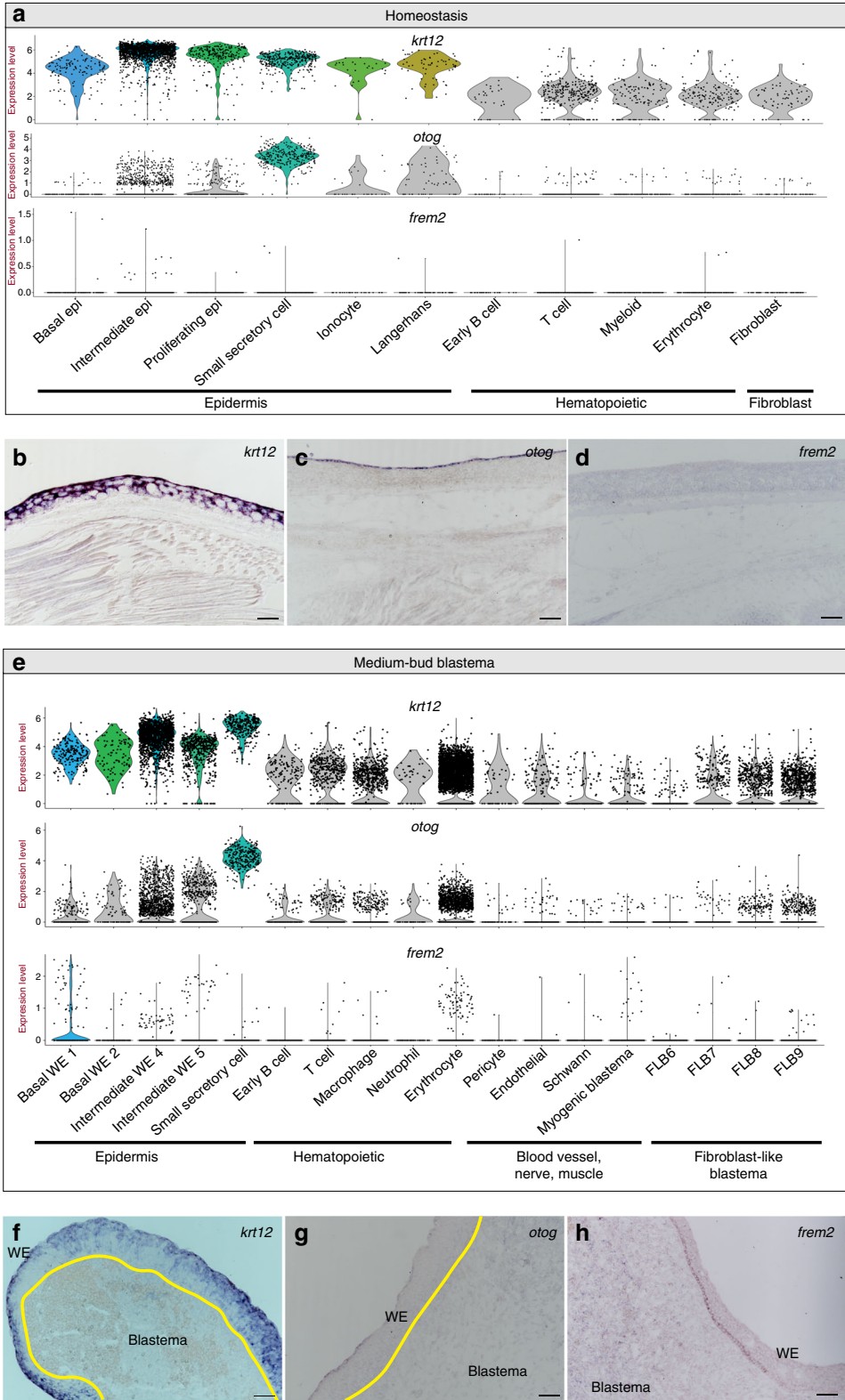

**Fig. 2** Wound epidermis gene expression dynamics during regeneration. **a** Violin plots depicting log$_2$ values of normalized expression of *krt12*, *otog*, and *frem2* at homeostasis. **b–d** Representative RNA in situ hybridizations in limbs at homeostasis probing for *krt12* (**b**) ($n = 5$ animals), *otog* (**c**) ($n = 3$ animals), and *frem2* (**d**) ($n = 3$ animals). **e** Violin plots depicting log$_2$ values of normalized expression of *krt12*, *otog*, and *frem2* in the medium-bud blastema. **f–h** Representative RNA in situ hybridizations in medium-bud blastemas probing for *krt12* (**f**) ($n = 5$ animals), *otog* (**g**) ($n = 3$ animals), and *frem2* (**h**) ($n = 5$ animals). In violin plots (**a**, **e**), each dot represents an individual cell; epidermal populations of interest are depicted in color, all others are in gray. Yellow line in **f**, **g** denotes boundary between the wound epidermis and blastema. Epi epidermis, WE wound epidermis, FLB fibroblast-like blastema cell. Magnification, ×10, scale bars are 100 μm

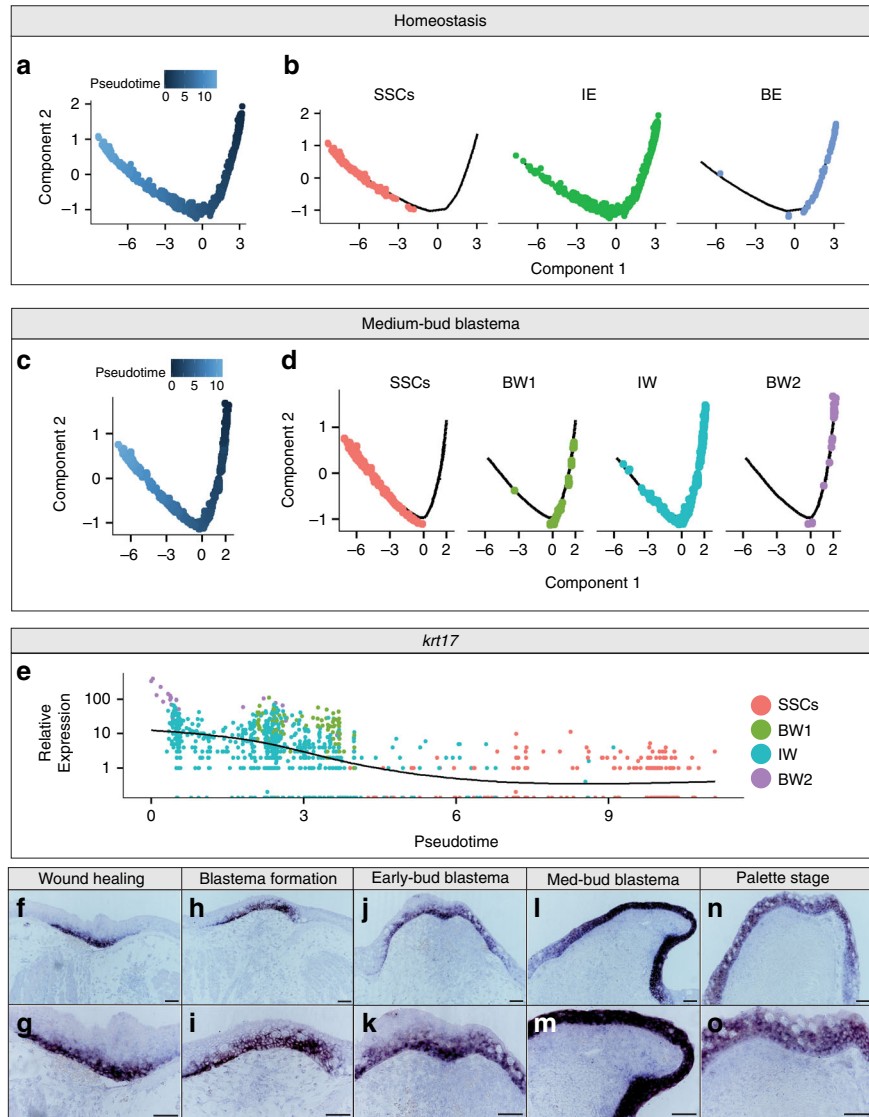

**Fig. 3** The epidermis follows an outward differentiating trajectory. **a–d** Pseudotime analysis of epidermis populations during homeostasis (**a**, **b**) and in the medium-bud blastema (**c**, **d**), shown split by overall pseudotime plots (**a**, **c**) and colored by Seurat population determined in Fig. 1 (**b**, **d**). Abbreviations used are small secretory cells (SSCs), intermediate epidermis (IE), basal epidermis (BE), basal wound epidermis #1 (BW1), intermediate wound epidermis (IW), and basal wound epidermis #2 (BW2). **e** Expression plot of *krt17* in epidermal cells captured at the medium-bud blastema stage over pseudotime, colored using Seurat populations as in **d**. **f–o** Representative RNA in situ hybridization probing for *krt17* across a time course of limb regeneration. Magnification ×10 in **f**, **h**, **j**, **l**, **n**, ×20 in **g**, **i**, **k**, **m**, **o**. All scale bars are 100 μm. $n = 4$ animals per time point

cells, which are likely from multiple epidermal populations. For both the homeostatic and regenerating epidermis, we observe an outward differentiation phenomenon starting at the basal epidermis/WE, which Monocle assigned as the earliest point in pseudotime (Fig. 3a–d). The populations we postulate to be intermediate epidermis served as intermediate populations connecting the basal epidermis to the outer layer of SSCs (Fig. 3b, d). Thus, at homeostasis and regeneration it appears that the basal epidermis likely serves as a progenitor cell reservoir.

Further we identify *keratin 17* (*krt17*) as a gene that was differentially expressed over pseudotime, with highest expression in the basal WE populations and expression tapering as cells entered intermediate and SSC populations (Fig. 3e). To confirm pseudotime analysis was capable of corroborating in situ gene expression profiles, we probed for *krt17* over a time course of limb regeneration. The expression profile of *krt17* over the course of limb regeneration verifies the correlation between pseudotime and real-time expression (Fig. 3f–o). Together, these data

highlight the cellular heterogeneity of the epidermis and WE, and will allow for the pursuit of the function of regeneration-induced genes.

**Adaptive and innate immune cells are present in the blastema**. It has been previously reported that hematopoietic-derived cells, in particular macrophages, play an important role in axolotl limb regeneration[13]. In addition, other myeloid cells are present in the regenerating limb, with infiltration peaking during wound healing and returning to baseline levels after blastema formation[13]. However, markers that are specifically tailored to identify axolotl immune cells have not been thoroughly investigated. This has hampered investigation of the role of immune cells and the factors they produce in limb regeneration.

Our analysis reveals distinct cell populations of both innate and adaptive immune cells are abundant throughout blastema formation. We find cells of the adaptive immune system, both T cells and B cells, are present at all time points sampled,

including medium-bud-stage blastemas (Supplementary Data 1–5). *Immunoglobulin lambda like polypeptide 5* (*igll5*) (Supplementary Fig. 3) and *t cell receptor alpha constant* (*trac*) (Fig. 4a) identified populations of B cells and T cells, respectively. Interestingly, *igll5* has been shown to be an early B cell marker[34], suggesting these cells may not be fully mature. At wound healing, we observe two populations of B cells, the second a presumably mature B cell population negative for *igll5*, but positive for B cell receptor machinery (Supplementary Fig. 3, Supplementary Data 3). We then identify *trac* as a gene expressed by T cells at all time points (Fig. 4a). To test if these adaptive immune cells were present in the blastema, we performed RNA in situ hybridization probing for *trac* to identify their location within the regenerate. We detect *trac*+ cells localized within the blastema (Fig. 4b–c). Previous work has implicated both CD8+ T cells and CD4+ regulatory T cells (T$_{regs}$) in murine muscle regeneration[35,36], and more recently, T$_{regs}$ have been shown to be required for spinal cord, heart, and retina regeneration in zebrafish[37]. However, whether these T cells in the blastema play a role in axolotl limb regeneration remains to be determined.

In addition to adaptive immune cells, we identify a plethora of innate immune cells present throughout the course of regeneration. These populations differ dramatically in the composition captured and molecular phenotype over the course of regeneration (Supplementary Data 1, 3–5). At homeostasis, beyond the Langerhans cells residing in the epidermis (Supplementary Fig. 1b), we also observe a population of putative myeloid cells, as defined by *macrophage expressed 1* (*mpeg1*) and *macrophage receptor with collagenous structure* (*marco*) expression[38,39] (Supplementary Data 2). We suspect that further investigation within this *mpeg1* and *marco* positive population may uncover both neutrophils and macrophages. In line with this, we observe both neutrophils and macrophages in medium- and early-bud blastemas (Supplementary Data 4–5).

As expected, we observe a dramatic increase in the myeloid cells captured and the complexity of the myeloid compartment during wound healing (Supplementary Data 3). We suspect these populations are a collection of macrophages, neutrophils, dendritic cells, and potentially other myeloid cells. Based on top markers of each population we propose that these cells are populations of dendritic cells expressing *runt-related transcription factor 3* (*runx3*) and *integrin subunit alpha e* (*itgae*) also known as CD103[40] (Supplementary Fig. 4a), newly recruited circulating macrophages expressing *triggering receptor expressed on myeloid cells 2* (*trem2*)[41] (Supplementary Fig. 4b), and three populations of neutrophils expressing *cathlecidin antimicrobial peptide* (*camp*)[42] (Supplementary Fig. 4c). If indeed the overwhelming proportions of leukocytes at this stage are neutrophils and recruited macrophages, our data point to an important role for myeloid cell recruitment to the amputation plane. However, this scenario would not rule out a role for tissue-resident myeloid cells contributing to limb regeneration.

Regardless of time point, myeloid cells could be identified by high expression of *apolipoprotein Eb* (*apoeb*) (Fig. 5a). We performed RNA in situ hybridization for *apoeb* across a time course of limb regeneration, which confirms that the presence of myeloid cells peaks during wound healing, and indicates that myeloid cells were present at all stages of regeneration sampled (Fig. 5b–g). Since previous reports in axolotl used α-naphthyl acetate (NSE) to detect myeloid cells in the blastema[13], we followed up the RNA in situ hybridization with NSE staining. As predicted by *apoeb* staining, we found that NSE detects myeloid cells within the medium-bud-stage blastema (Supplementary Fig. 5a–b). Many of these cells lack canonical markers used to identify immune cells in other species. However, cellular morphology is likely conserved and lymphocytes, neutrophils,

macrophages, eosinophils, and mast cells have previously been described histologically in the newt blastema[43]. Together, our data provide integral information about the identity of axolotl immune cells, which will allow for further study of their contribution to limb regeneration.

**Cellular sources of nerve-associated cells and blood vessels**. Re-innervation and re-vascularization are essential processes during limb regeneration, requiring the orchestration of re-growing nerves and vessels in coordination with supportive cell types. Within our dataset, we find populations of putative Schwann cells, pericytes, and endothelial cells that may contribute to re-innervation and re-vascularization of the regenerating limb. Endothelial cells, as determined by specific *platelet and endothelial cell adhesion molecule 1* (*pecam1*) expression could be found at all three regenerating time points (Supplementary Fig. 6a). We identify pericytes through expression of markers such as *cytoglobin* (*cygb*)[44] and *atp-binding cassette subfamily c member 9* (*abcc9*)[45] (Supplementary Fig. 6b). We only detect a robust number of pericytes in the medium-bud blastema, in accordance with the delayed arrival of pericytes reported in axolotl digit tip regeneration[4]. As for nerve-associated cells, we find cells during wound healing and in medium-bud blastemas expressing *myelin protein zero* (*mpz*), a well-described Schwann cell marker[46] (Supplementary Fig. 6c). Interestingly, during wound healing, Schwann cells were mainly negative for *sry-box 2* (*sox2*); however, in the medium-bud blastema, many of these cells had acquired *sox2* expression, which may serve to prevent myelination and maintain Schwann cells in a non-differentiated state[46] (Supplementary Fig. 6c). In addition, we observe the presence of the neural crest marker *forkhead box d3* (*foxd3*)[47] in the medium-bud blastema Schwann cells, furthering the notion that these cells have likely entered a less differentiated state (Supplementary Fig. 6c).

**Single-cell RNA-seq elucidates blastema cell identity**. The literature suggests that Pax7+ muscle satellite cells, dermal fibroblasts, and cells residing within the periosteum, likely also fibroblasts, contribute to the mesenchymal components of the blastema[4–6,15,16]. Based on these data we reasoned that there would be a population of blastema cells derived from Pax7+ progenitors that were responsible for muscle regeneration[6,16]. Consistent with this hypothesis, we observe populations of cells in early- and medium-bud blastemas that specifically express *paired box 7* (*pax7*) and *myogenic factor 5* (*myf5*) (Supplementary Fig. 6d–e). However, the expression of these two markers is inversely correlated, with presumptive myogenic blastema cells in the early-bud blastema expressing *pax7* without robust *myf5*, and the converse in the medium-bud blastema. These data indicate more consistent/sustained markers for myogenic blastema cells may be *six homeobox 1* (*six1*) and *eya transcriptional coactivator and phosphatase 1* (*eya1*), which mark this population at both time points sampled (Supplementary Fig. 6f–g). In mice, both *six1* and *eya1* are essential transcription factors for muscle development[48,49]. While the converse nature of *pax7* and *myf5* expression may hint at different states of differentiation, it is also possible that the low read coverage of microfluidic-based single-cell RNA-seq makes capturing and detecting lowly expressed transcription factors (i.e., potentially *pax7*) difficult[21]. It should be noted that these presumptive myogenic progenitor cells are not obviously detected until the early-bud blastema, which is concordant with the apparent delayed arrival of muscle progenitors into the blastema[50].

Next, we looked for populations of cells that appeared to be similar to fibroblasts. We observe populations of cells at all time

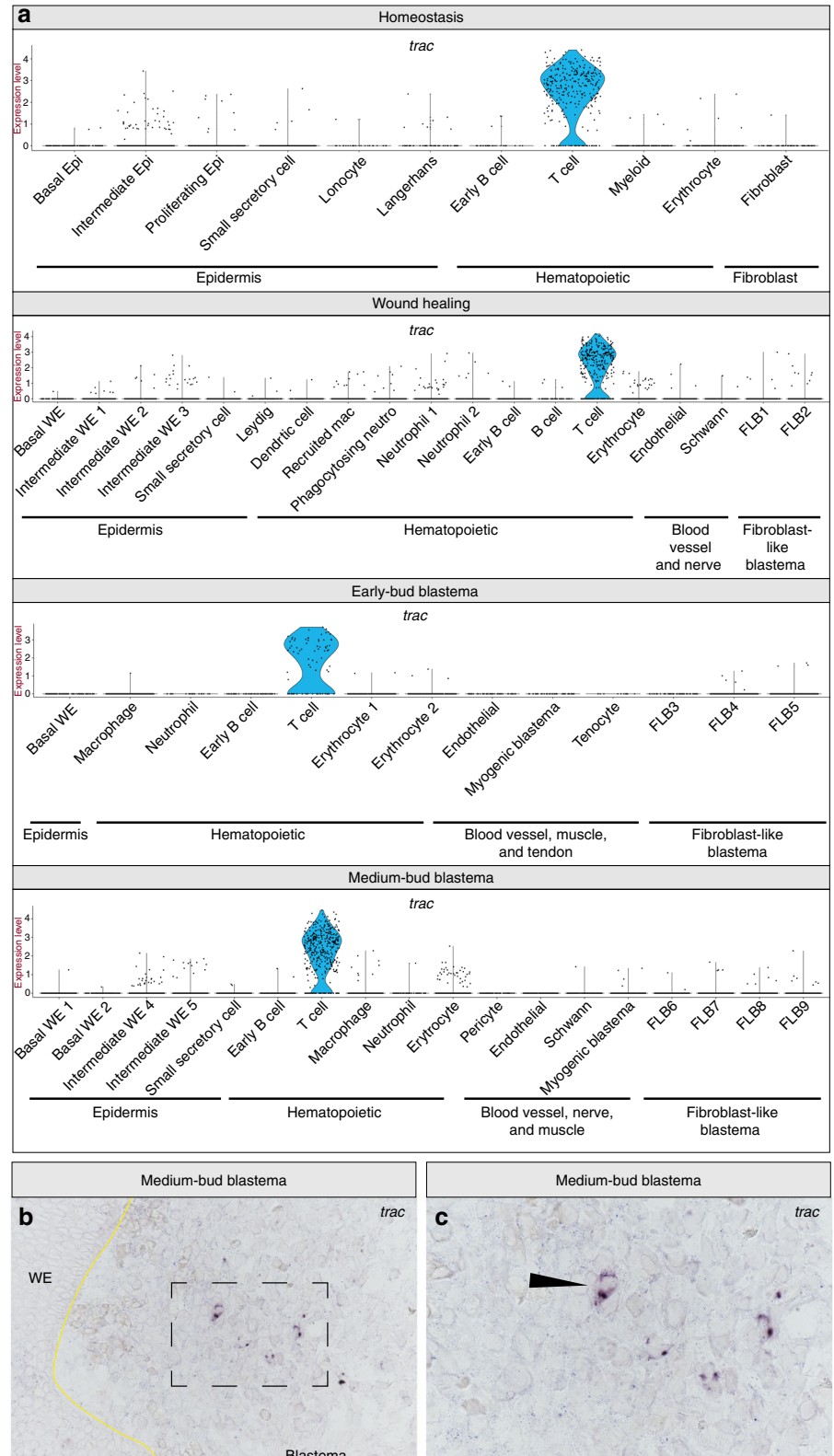

**Fig. 4** The blastema contains T cells. **a** Violin plots depicting log$_2$ values of normalized expression of the T cell-specific gene, *trac*, at homeostasis, wound healing, early-bud blastema and medium-bud blastema stages. Each dot represents an individual cell. **b**, **c** Representative RNA in situ hybridization probing for *trac* in a medium-bud-stage regenerate. **b** Magnification, ×20 scale bar is 100 μm; **c** area outlined in **b** at ×40 magnification, scale bar is 50 μm. $n = 4$ animals

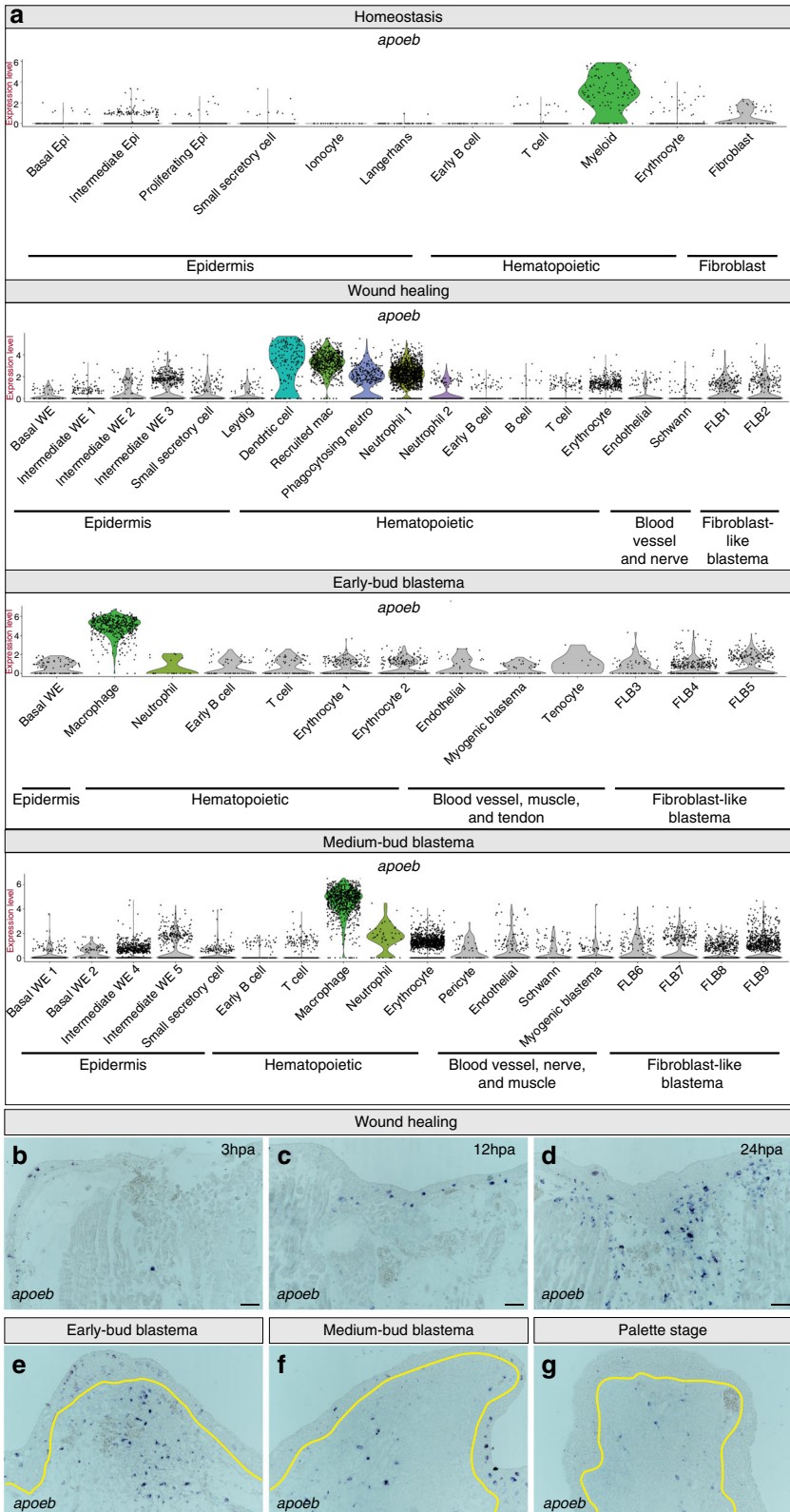

**Fig. 5** Myeloid cells are present throughout the course of regeneration. **a** Violin plots depicting $\log_2$ values of normalized expression of myeloid cell enriched *apoeb*. Each dot represents an individual cell; myeloid cell populations are in color, while all other populations are in gray. **b**–**g** Representative RNA in situ hybridizations probing for *apoeb* across a time course of regeneration. **b**–**g** Magnification ×10 during wound healing, specifically at **b** 3 h post amputation (hpa), **c** 12hpa, and **d** 24hpa. **e**–**g** Magnification ×10 of a **e** early-bud blastema, **f** medium-bud blastema, and a **g** palette stage-regenerating limb; yellow line denotes border of blastema and WE. All scale bars are 100 μm. $n = 4$ animals per time point

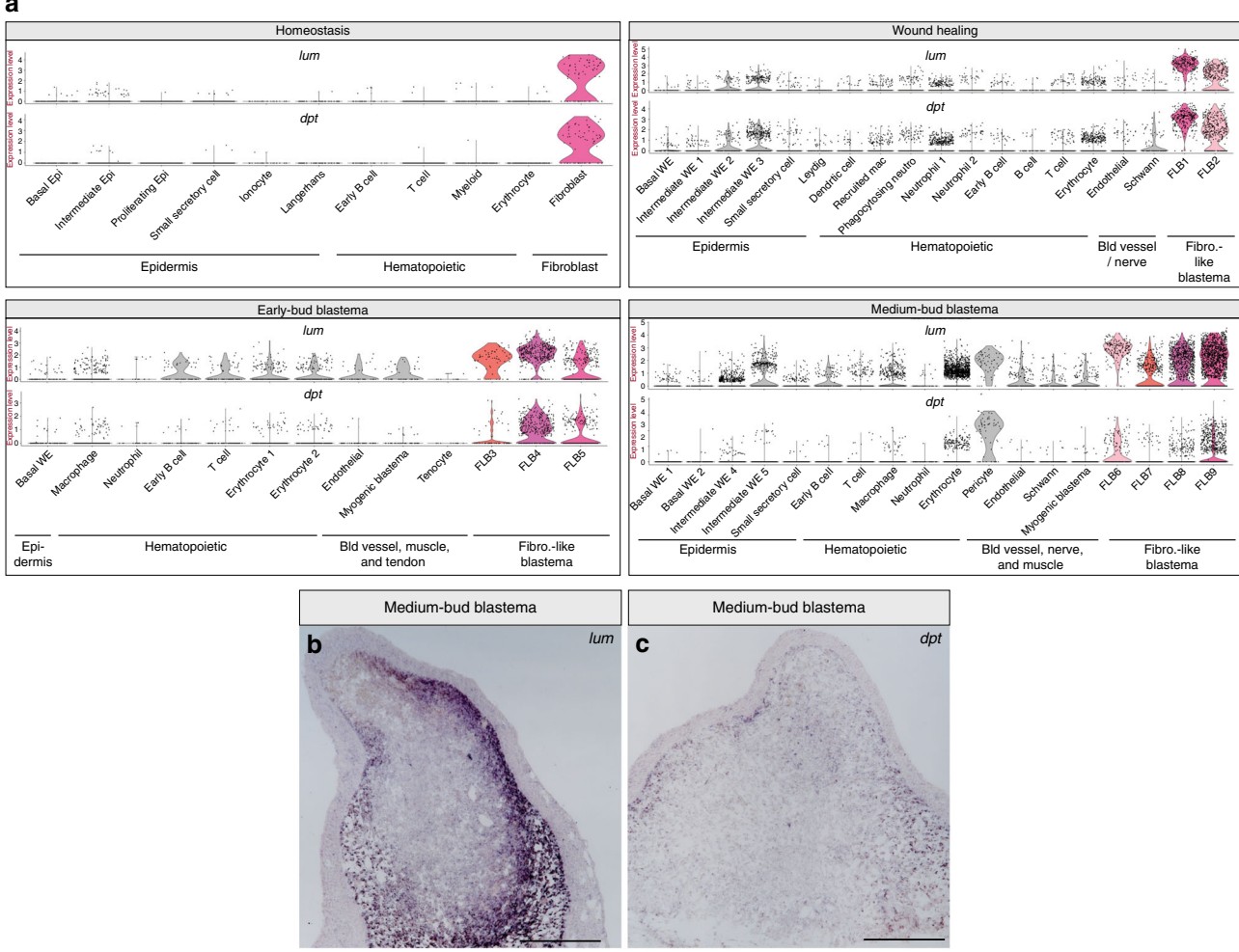

**Fig. 6** Fibroblasts-like cells are found during homeostasis and regeneration. **a** Violin plots depicting log$_2$ values of normalized expression of fibroblast and fibroblast-like blastema (FLB) cell-enriched genes *lum* and *dpt* at homeostasis, wound healing, early-bud blastema, and medium-bud blastema stages. Each dot represents an individual cell; fibroblast/FLB populations of interest are depicted in color, while all other populations are in gray. **b** Representative RNA in situ hybridization probing for *lum* in a medium-bud blastema. **c** Representative RNA in situ hybridization probing for *dpt* in a medium-bud blastema. Magnification ×4, scale bar is 500 μm. $n = 4$ animals per probe

points, including at homeostasis, that appear to be fibroblast-like cells. These fibroblast-like cells have high expression of *lumican* (*lum*), *dermatopontin* (*dpt*), and *periostin* (*postn*), which in mice are expressed by fibroblasts and periosteal cells[51–53] (Fig. 6a, Supplementary Data 3–5). We note that pericytes also express *lum* and *dpt* at the medium-bud-stage, suggesting that these cells may acquire some fibroblast-like markers (Fig. 6a). We refer to these fibroblast-like cells as FLB cells and numbered the populations by time point to facilitate later identification: FLB cells 1–2 present at wound healing, FLB cells 3–5 present in the early-bud blastema, and FLB cells 6–9 in the medium-bud blastema. To confirm the presence of these FLBs, we performed RNA in situ hybridization on medium-bud blastemas for *dpt* and *lum* and find that these markers are expressed in distinct populations of the medium-bud blastema (Fig. 6b, c).

To begin to understand how the different blastema cell populations may be contributing to the overall regenerative landscape, we attempted to delineate the potential differentiation trajectories of these cells. We focused on cells that would potentially contribute to the structure of the regenerated limb, including FLB cells, Schwann cells, myogenic blastema, endothelial cells, and pericytes. To determine potential relationships in these populations across different points in regeneration we used

URD, an algorithm that was used to reconstruct developmental trajectories in the developing zebrafish[54]. We performed unbiased clustering of the aforementioned blastema-resident cells from the three regenerating time points. As expected, these cells cluster based on time point sampled, highlighting the differences in transcriptional states during regeneration (Fig. 7a). We selected cells from wound healing as root populations, reasoning that these cells would give rise to progeny in the blastema at later stages.

We then calculated pseudotime ordering assuming that the root populations were at pseudotime zero. We find that cells present in both the early- and medium-bud blastema exhibit bimodal pseudotimes, suggesting that both time points had cells at earlier and later stages of differentiation (Fig. 7b). We then attempted to identify cells in the medium-bud blastema that could serve as terminal cell populations in the regenerative trajectory. Since these cells are likely still located along a trajectory towards a truly terminal cell population, we refer to these tip populations as trajectories. We identified 10 populations within the medium-bud blastema, which based on marker gene expression could be endothelial cells (*pecam1*), Schwann cells (*sox10*), pericytes (*cygb*), myogenic blastema (*myf5*), fibro-adipogenic progenitors (FAPs) (*pdgfra*, *osr1*[55]), a population of

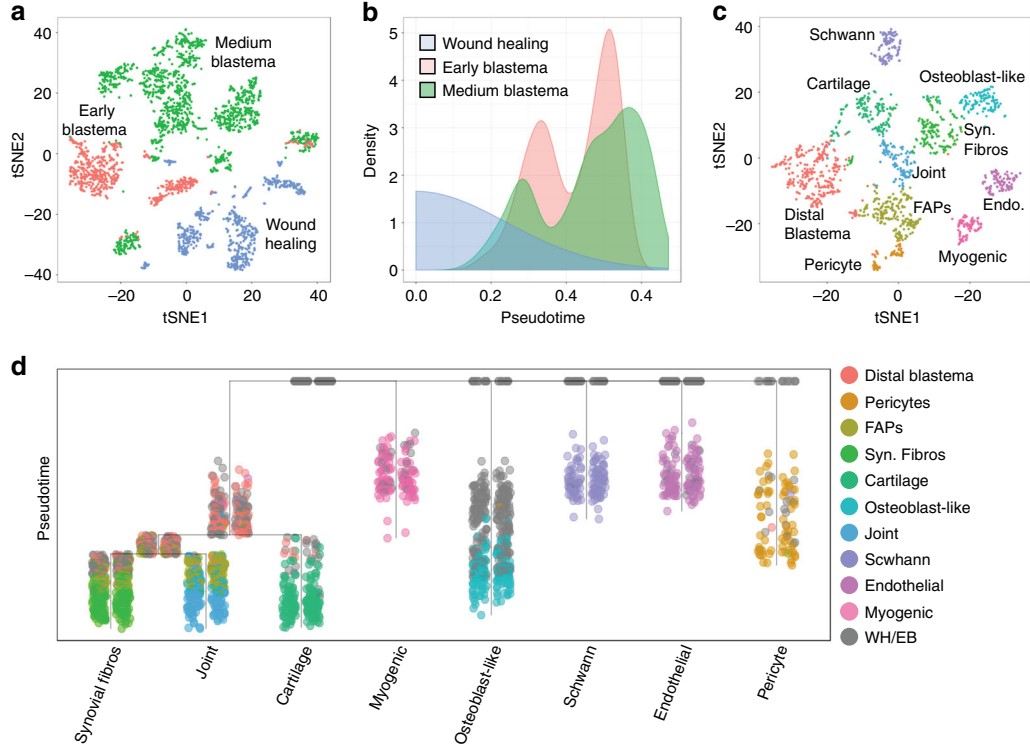

**Fig. 7** Putative differentiation trajectories of blastema cells. **a** t-SNE plot of blastema cells colored by respective time point sampled. **b** Distribution of pseudotime of blastema cells colored by time point sampled. **c** t-SNE plot of blastema cell populations in the medium-bud-stage blastema. **d** Blastema differentiation trajectories colored by blastema cell populations identified in **c**. Syn. fibros synovial fibroblasts, Endo. endothelial cells, FAPs fibro-adipogenic progenitors, WH wound healing, EB early-bud blastema

undifferentiated FLB cells with distal markers (*hoxa13*[56]), and fibroblasts on trajectories towards osteoblast-like cells (*bglap*[57], *odam*[58]), joint-like (*gdf5*[59]*barx1*[60]), cartilage (*col2a1*[25]), and synovial fibroblasts (*cdh11*[61]) (Fig. 7c, Supplementary Data 6–15).

We reasoned that these cells, except for FAPs and undifferentiated distal blastema cells, were potentially distinct trajectories. We then determined potential relationships by plotting a tree using the aforementioned blastema-resident cells from wound healing, early-bud, and medium-bud blastema stages. These data suggest that synovial fibroblasts, joint-like and cartilage trajectories potentially share a common fibroblast-like progenitor (i.e. distal undifferentiated blastema) (Fig. 7d). In contrast, pericytes, endothelial cells, Schwann cells, myogenic blastema cells, and osteoblast-like fibroblasts have distinct trajectories (Fig. 7d). In addition, FAPs lie along the joint-like and synovial fibroblast-like trajectories, suggesting that these cells are potentially an intermediate cell state between undifferentiated blastema cells and these tips (Fig. 7d). Notably, it appears that putative osteoblast-like fibroblasts are a distinct lineage compared to the other fibroblasts in this analysis. This suggests the potential for different cellular origins of FLB cells, and that tissue-resident fibroblasts at homeostasis may be a heterogeneous population. However, lineage-tracing experiments would be required to confirm this hypothesis. We plotted markers that defined the tip populations to confirm the potential tissue trajectory of each population (Supplementary Fig. 7). By overlying gene expression on the trajectory tree, it is clear that some of these marker genes may be expressed at different stages of differentiation. Further exploration of factors that define branch points may lead to information about cell state transitions and decisions in the multipotent undifferentiated blastema cells.

In the above analysis we did not include a rare population of cells (11 total) only found in the early-bud blastema. This population was marked by expression of *scleraxis*, which is a definitive marker for tenocytes[62] (Supplementary Data 4). Interestingly, this putative tenocyte population did not have expression of fibroblast-associated markers, suggesting either a non-fibroblast origin of these tenocytes or a complete de-differentiation process to a progenitor population. Thus, the main components of the regenerated limb likely derive from a diverse array of origins, including, but not limited to, rare tissue-resident progenitor cell populations (i.e. muscle satellite cells and potentially tenocytes) and de-differentiation and subsequent re-differentiation of fibroblasts and Schwann cells. Overall, these data suggest that fibroblasts-like cells can form different types of tissue and that the other lineages we uncovered appear to be fate-restricted by their original identity.

## Discussion

These data provide an important, detailed molecular resource that outlines and defines the cell populations that contribute to axolotl limb regeneration. In particular, this study provides markers that separate unique populations during regeneration and at home-ostasis within the epidermis, blastema progenitors, and hematopoietic cells. We highlight the populations that have previously been described to play integral roles in regeneration, but in general have lacked more comprehensive molecular descriptions. This includes the basal WE, macrophages, FLB cells, and Pax7+ muscle satellite cells. However, these data show that a variety of other cell populations are present during regeneration, and we suspect that clues to how regeneration occurs may be found in many of these populations. It will be important to remain open to the possibility of unexpected or rare cell populations playing fundamental roles during regeneration.

There are contrasting thoughts on how the axolotl limb is regenerated, whether all populations are derived from rare progenitors resident in the homeostatic limb or if de-differentiation of stump-resident cells provides a reservoir of progenitor cells. Our data give credence to both hypotheses. To support the rare progenitor hypothesis, we uncovered previously described Pax7+ muscle satellite cells[6,16]. This hypothesis was further supported by our discovery of a rare population of putative tenocytes that appears to not be fibroblast-like (Supplementary Data 4). This suggests the regenerating tendon may form from a rare tissue-resident progenitor like that seen in mice[63]. In support of the de-differentiation hypothesis, our data suggest that Schwann cells may return to a precursor-like state, acquiring *sox2* expression (Supplementary Fig. 6c), which is a feature of de-differentiated Schwann cell precursors observed in mouse digit tip regeneration[64]. Further investigation as to whether, similar to mice, this population provides essential growth factors and is thus required for limb regeneration is warranted.

While our data suggest that FLB cells are capable of multi-lineage potential (Fig. 7d), it is not clear to what extent the formation of these cells is due to either recruitment of rare tissue-resident progenitors or de-differentiation of fibroblasts. Importantly, we did capture a population of fibroblasts within the homeostatic limb that closely resembled FLB cells (Fig. 6, Supplementary Data 2–5), though the limited number makes it difficult to draw broad conclusions about this population. These fibroblasts likely represent dermal fibroblasts and periosteal cells within the homeostatic limb. Capturing more of these homeostatic fibroblasts may elucidate rare progenitor populations or heterogeneity that suggests a bias towards regenerating into a certain tissue type. Similarly, the osteoblast-like fibroblasts we detect in the medium-bud blastema may be derived from a separate cell population even during wound healing (Fig. 7d). Future experiments with lineage-tracing tools will be required to elucidate what cells transition from the stump to form the blastema. In general, it appears that fibroblast-like cells, in particular the population of undifferentiated blastema, have the highest degree of multipotency (Fig. 7d), which is consistent with previous work[3,4]. Therefore, while it appears that cells enter a blastema-like state via variety of mechanisms, it is now more clear that fibroblasts are multipotent, likely forming the joints, cartilage, and bone of the regenerate. In contrast, blastema cells contributing to tendon, muscle, blood vessel, and nerve may be lineage-restricted.

While understanding how the blastema forms is of utmost importance, regeneration cannot proceed without the contributions of other cell types. For example, the WE is required for limb regeneration, yet a detailed understanding of the basal WE, which directly overlies the blastema mesenchyme, has remained elusive. These data provide markers for future lineage-tracing experiments, as well as a glimpse at transcripts that are dynamically regulated along the course of regeneration. We highlight one such marker, *frem2* (Fig. 2), which has an essential role in tissue morphogenesis during development[65]. Targeting the function of genes enriched in the basal WE at different time points will provide fundamental knowledge about contributions of this population to regeneration.

These data further highlight the potential influence of immune cells on limb regeneration and raise the intriguing and unexplored possibility that the adaptive immune system may be a player in limb regeneration (Fig. 4, Supplementary Fig. 3). This is reinforced by a recent study in zebrafish showing a fundamental role for T$_{regs}$ in the regeneration of a variety of organs[37]. Since these data describe the landscape of the blastema niche, and not blastema progenitors alone, it will be important to investigate contributions from other cell types (i.e. immune cells, WE) that may help guide blastema progenitor cells to a pro-regenerative state.

Pre-existing RNA-seq and microarrays resources built from different tissues and experimental setups in axolotl limb regeneration have aimed to identify genes that have dynamic expression during limb regeneration[17]. Teasing out the contribution of transcripts expressed in the blastema versus WE, as well as the diversity of cell populations that we have now shown to be present in the blastema, is critical. These single-cell RNA-seq data also provide a new lens to re-evaluate bulk-tissue experiments. While single-cell RNA-seq has proved to be a revolutionary technology, it is still a developing technique, and issues of cell capture efficiency (which can vary for different shaped/sized cells), transcript capture efficiency, and read depth may limit current studies. Notably, we did not uncover some previously described blastema-enriched genes, such as *kazald1*[66], at appreciable levels in our dataset, likely due to one of these technical limitations or due to still-improving transcriptomic and genomic assemblies. Further, computational methods to successfully define populations and subpopulations are rapidly improving, and revisiting datasets in the future may provide a more nuanced view of current studies. With the recent transcriptional[17] and genomic[16,67] resources now available for axolotl, continuing to annotate the genome and generate a consensus transcriptome will be imperative for unifying studies of this important model system.

In general, many of the populations for which we present detailed molecular-level information appear to be entering a developmental-like state. The reuse of transcription factors such as *six1*, *eya1*, *scx*, *sox2*, and *foxd3* strongly suggests that limb regeneration recapitulates development, which would be an efficient strategy even if there are distinct considerations including inflammation, injury site variance, differences in target limb size, and many others. This adds further intrigue to the question as to why the ability to re-initiate a developmental-like program to regenerate a limb is limited to so few species. Since more is known about limb development than limb regeneration, and species such as humans already possess the capacity to develop limbs, more research focused on differences between these two processes is warranted. Overall, these data provide molecular-level detail on the cellular composition of the blastema niche, which enhances our understanding of this complex structure.

## Methods

**Animal experimentation**. All experiments using axolotls were performed in accordance with the Brigham and Women's Hospital Institutional Animal Care and Use Committee under protocol #2016N000369. For all amputations, animals were narcotized in 0.1% tricaine and subsequently kept in 0.5% sulfamerazine overnight. All single-cell RNA-seq experiments were performed on male and female axolotls over the age of 1 year with lengths between 17 and 21.5 cm snout to tail. All animals were never previously amputated and were housed singly their entire lives to prevent cannibalistic biting of limbs.

**Single-cell collection and sequencing**. A detailed description of the cell harvest protocol can be found here (dx.doi.org/10.17504/protocols.io.qmmdu46). This protocol was derived with modifications from previous publications[3,68,69]. This protocol consistently provided both blastema and epidermal cells, but was biased towards epidermal collection. Therefore, due to small blastema cell quantity in the early-bud blastemas, the entire WE was removed, and for medium-bud blastemas, the WE was removed and only half of the WE was put into the collection tube. Cells were then run on inDrops and libraries of about 3000 cells were generated[21,22] at the Single Cell Core at Harvard Medical School. Sequencing was performed at the Dana-Farber Cancer Institute Molecular Biology Core Facility using the inDrops V3 design, on a NextSeq500 (Illumina) using the following parameters: read type = paired-end, cycles read1 = 61, cycles read2 = 14, indexing = DualIndex, cycles index 1 = 8, and cycles index 2 = 8.

**InDrops data processing and expression profiling**. RNA-seq data were demultiplexed using the inDrops processing pipeline (https://github.com/indrops/indrops) which was forked (https://github.com/brianjohnhaas/indrops) and

modified for use with a previously published axolotl transcriptome (http://portals.broadinstitute.org/axolotlomics/)[66]. Reads, sample identity, cell barcodes, and unique molecular identifiers (UMIs) were extracted from the Illumina data and encoded into a single fastq file. Reads were aligned to the axolotl transcriptome using bowtie like so: bowtie Axo.Trin.fasta.bowtie -q -p 20 -a --best --strata --chunkmbs 1000 --sam -m 200 -n 1 -l 15 -e 100 reads.fastq > alignments.sam.

Aligned reads were counted according to targeted transcripts (using script 'bam_to_count_matrix.pl'), with reads fractionally assigned across multi-mapped targets, restricted to unique UMIs at target transcripts. Finally, read counts were aggregated across transcript isoforms to yield a gene count matrix, in which a single representative isoform was chosen to represent each respective gene.

**Unbiased clustering with Seurat**. Cell by gene matrices were filtered to retain only genes that were found in at least eight cells in all samples. Data were imported into R and analyzed using Seurat 2.3.4[23]. Analysis was conducted as described (http://satijalab.org/seurat/) with modifications. Cells with low numbers of genes or high mitochondrial RNA expression were removed. Variable genes for each time point were determined, data were scaled, and linear dimensional reduction was performed. Principal components used for clustering were determined by visual inspection of a plot of standard deviations of the principal components. To determine cluster number the data were over-clustered (i.e. a high resolution was used that separates out populations that share highly similar gene expression profiles). This was followed by examination of cluster distinctiveness using a random forest classifier, and subsequent merging of populations that scored highly (i.e. were highly similar). All code used in in this paper can be found at https://github.com/brianjohnhaas/indrops.

**Pseudotime analysis with Monocle**. Using Monocle 2.6.4, Seurat objects were imported and converted directly into Monocle CDS objects. After subsetting the epidermis populations of interest, the remaining cells were clustered, and differentially expressed genes were determined. Following dimensional reduction using the DDRTree method, cells were ordered along a trajectory. Analyses were based on Monocle documentation (http://cole-trapnell-lab.github.io/monocle-release/docs/). Full R markdown of this analysis can be found at https://github.com/brianjohnhaas/indrops.

**Trajectory analysis with URD**. All non-immune cells residing within the blastema cells (FLB, myogenic blastema, Schwann cells, endothelial cells, and pericytes) were extracted from wound healing, early-bud blastema and medium-bud blastema time points. Tenocytes were removed due to the small cell number (11 cells). As per the URD tutorial (https://github.com/farrellja/URD), transition probabilities and pseudotime were calculated and potential tip populations were determined and refined. Biased random walks were then performed to determine potential relationships between cell populations. Detailed code on URD implementation can be found at https://github.com/brianjohnhaas/indrops.

**Code availability**. All code/details used to process raw data to cell by gene matrices in this manuscript can be accessed at https://github.com/brianjohnhaas/indrops along with R code which is also available as Supplementary Code.

**Statistics**. Differentially expressed genes between Seurat determined clusters were evaluated using the non-parametric Wilcoxon's rank-sum test and adjusted $P$ values were calculated using Bonferroni correction. For URD clustering an area under the precision-recall curve was used to determine cluster-enriched genes. All RNA in situ hybridizations were performed a minimum of three times. Single-cell RNA-seq was performed in three separate experiments/days. Medium-bud blastema samples were acquired in two sets of three on different days. One set of medium-bud blastema was processed with two early-bud samples, and the other with three wound healing samples. The homeostatic limbs were collected in a third separate experiment.

**Specimen harvesting and processing**. Regenerating limb tissue was collected and immediately fixed in 4% paraformaldehyde in diethyl pyrocarbonate-treated 1× phosphate-buffered saline (PBS) for 1–2 h at room temperature, washed in 1× PBS, and transferred through a sucrose gradient up to 30%. Tissue was then cryopreserved into molds using TissueTek Optimal Cutting Temperature compound. Tissue was sectioned at 16 μm on a Leica cryostat and stored at −80 °C prior to experimentation.

**RNA in situ hybridization**. Probes were amplified from regenerating limb complementary DNA and cloned into pGEM-T-easy and sequenced. The primers used for generating all probes used in this study can be found in Supplementary Table 2. Anti-sense probe was generated with either T7 or Sp6 polymerase depending on orientation in pGEM. Tissue was collected at stages specified according to Tank et al.[70] from animals that were 18–22 cm in length and processed as described above. For time-course sections, animals were 9.5–11.5 cm and harvested at stages indicated. A detailed step-by-step RNA in situ hybridization protocol used in this paper can be found at protocols.io (dx.doi.org/10.17504/protocols.io.p33dqqn)

**NSE staining**. After harvesting and sectioning as described above, tissue was post-fixed in 4% paraformaldehyde for 10 min, rinsed with running $dH_2O$ for 2 min, and then incubated in NSE stain for 10 min at 37 °C. Slides were then rinsed for 2 min in $dH_2O$ before coverslipping.

**Hematoxylin and eosin staining**. After harvesting and sectioning as described above, tissue was rehydrated in 1× PBS, post-fixed in 4% paraformaldehyde in 1× PBS for 10 min, rinsed with 1× PBS, and then water. Hematoxylin and eosin staining was then performed in buckets as follows: xylene for 5 min (×2), 100% ethanol for 2 min (×6), water for 2 min, hematoxylin for 3 min, water for 2 min, acid alcohol for two quick dips, water for 2 min, Scott's blueing agent for 3 min, water for 2 min, eosin for 2 min, 100% ethanol for 2 min (×5), and xylene for 2 min (×2). Coverslips were then applied and slides were left to dry overnight.

## Data availability

The authors declare that all data supporting the findings of this study are available within the article and its Supplementary information files or from the corresponding author upon reasonable request. All single-cell RNA-seq data and cell by gene matrices used to generate all graphs in this manuscript have been deposited in the Gene Expression Omnibus database under accession code GSE121737. A reporting summary for this Article is available as a Supplementary Information file.

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

## Acknowledgements

This work was supported by the Eunice Kennedy Shriver National Institute of Child and Human Development (DP2HD087953 to J.L.W.) and the National Institute of Arthritis and Musculoskeletal and Skin diseases (R03AR068126 to J.L.W.) of the National Institutes of Health and the Richard and Susan Smith Family Foundation (J.L.W.). N.D.L. was supported by Award Number F32HD092120 from the Eunice Kennedy Shriver National Institute of Child and Human Development of the National Institutes of Health. D.M.B. was supported by a HHMI Gilliam Fellowship. R.M. was supported by the National Science Foundation Graduate Research Fellowship (DGE1144152). R.O. was supported by PRISE (Harvard University) and Herchel-Smith (Harvard University). A.Y. W. was supported by HCRP (Harvard College) and HIP (Harvard Stem Cell Institute). We also thank the Harvard Medical School Single Cell Core for inDrops cell collection; Dana-Farber Cancer Institute Molecular Biology Core Facilities for sequencing; Adam

Gramy and Sarah Lemire for help with animal care; Tae J. Lee and Duygu Payzin-Dogru for discussion and comments.

## Author contributions

N.D.L., R.M., and J.L.W. designed experiments; N.D.L., and J.L.W. wrote the paper; N.D.L, R.M, G.S.D., A.C., and B.J.H. performed computational analyses; N.D.L., A.R., R.M., K.J., R.O., A.Y.W., D.M.B, W.W.Y., and B.M.M. performed sample preparations and validations. All the authors contributed to manuscript editing.

## Additional information

**Competing interests:** The authors declare no competing interests.

