## [Peer Review File · Nature Communications]

Reviewers' Comments:

Reviewer #1:

Remarks to the Author:

This paper explores the cellular composition of the regenerating axolotl limb using a single cell RNAseq approach, which is complemented by in situ hybridizations with probes against a number of selected transcripts. Determining the extent of heterogeneity of the axolotl limb during regeneration is an important and timely aim. However, the manuscript is preliminary at this stage and does not live up to its rather pompous title. In contrast to the authors, I would argue the paper does not provide "a rich resource.....for axolotl limb regeneration". As I see it, the main problem is the very low representation of mesenchymal non-blood blastemal cells in the sample, which appears to be well below 10% (I did not find the exact figure). The blastema is the origin of the regenerating limb but this low number of sequenced cells does not allow to generate a proper representation of its composition. The conclusion that the blastemal cells are "highly homogenous" is not well supported by the low number of sequenced blastemal cells. Thus, the progress in our understanding of limb regeneration provided by the paper is quite limited in that sense. The strong part of the paper is the characterization of the wound epidermis, since the vast majority of RNAseq data are derived from that tissue. While the wound epidermis data are solid, they don't provide significantly new insights into the biology of the wound epidermis. The finding that the wound epidermis can be subdivided is not new (see Tassava, 2000, which the authors also refer to) even though the authors provide additional markers for the previously defined regions. The analyses of myeloid and cells and lymphocytes is relevant but here the validation of the RNAseq data is weak. I am also surprised how the authors used GO terms. GO terms are often unpecific and the authors highlight particularly broad terms such as "negative regulation of biological processes". A way forward could be to generate more RNAseq data, particularly of blastemal cells, or to carry out functional studies with selected candidates and use the single cell data as an entry point to these studies.

Reviewer #2:

Remarks to the Author:

This paper describes a study wherein single-cell RNA-seq was performed on 3500 cells collected from 3 medium-bud blastemal staged limbs of two adult axolotls. This is an important advance for the axolotl model and regenerative biology more generally as studies have primarily used whole tissues for gene expression analyses of regeneration. The work is original, timely and broadly interesting, and is likely to motivate similar studies in the field. My comments below are meant to improve the presentation of results from this important study.

1) The manuscript could benefit from a bit more consistency in terminology. In the introduction, the paper uses "cell types" to describe cells that were discovered by RNA-seq to exhibit the same/similar pattern of gene expression. However, in the results, the paper uses "cell populations". Because the study only sampled a single developmental time point within the regeneration process, from only three limbs, it is probably more accurate to refer to the different cell clusters as populations. As another example, "unbiased clustering" and "unsupervised clustering" are used to describe the method for identifying cell populations. At some point in the paper it would be helpful to briefly state what is meant by unbiased clustering. Does this apply to the algorithm used to define clusters as well as the method used to determine cluster number?

2) Several of the Gene Ontology terms that were identified for cell populations are highly inclusive of many biological processes and care should be used in their interpretation. On lines 108-110, GO: Regulation of Biological Process does not necessarily equate to regulation of growth and proliferation, and GO: Multicellular Organism Process necessarily equate to tissue and appendage morphogenesis. There are other examples in the paper.

3) One of the genes discovered as a marker of intermediate WE (pinlyp) was recently shown to be highly differentially expressed among replicate samples during axolotl limb regeneration. Did this gene show such a pattern in this study and is there concern that some genes may exhibit bursting or on/off phenotypes among cells when performing single-cell analyses? I raise this issue because there is no general discussion of caveats (excepting Line 377 where the importance of a single cell phenotype is discussed, which isn't really worth discussing) to single-cell sequencing of a moderate number of cells from few samples. Seems like cell number, sequencing depth, dealing with a large, unannotated genome, etc, should deserve a bit more attention.

4) The paper could make a more valuable contribution to the axolotl and more generally amphibian models of regeneration if there was more integration of information about the biology of amphibian skin. There are in depth histological descriptions of amphibian skin and cell types (some of these are outside of the regeneration field but applicable, even considering differences in wound epidermis and normal epidermis). I see no reason to awkwardly introduce an interest in genes expressed during mouse development or during regeneration of gerbil skin (lines 142-145), or to focus on mammalian stem cells (Lines 292-293) when such cells are known for amphibian skin.

5) Lines 202-206 offer a "rapid evolution" explanation for not identifying gene ontologies for the myeloid population. This seems a bit of a stretch. If orthologs can be identified, then appropriate DNA sequence comparison tests could be performed. If many of these genes are unannotated/anonymous, then perhaps write that the number of unannotated/anonymous genes for this cell population is higher than for other cell populations.

6) In the section describing T cells, it would be good to cite the recent paper by Hui et al in Developmental Cell showing the requirement of Tregs in multiple examples of zebrafish organ regeneration.

7) Line 282: Probably the word "diversity" is better than "contamination".

8) Line 320: Macrophages have an early role in regeneration but their activities do not initiate regeneration.

9) Line 351 suggests that the single cell data presented in the paper provides a new lens to re-evaluate bulk tissue experiments. It would improve the paper if this were addressed explicitly with examples that concern estimates of gene expression and not issues concerning gene orthology, which is a separate issue. For example, a recent transcriptome analysis of the axolotl blastema (Bryant et al in Cell Reports) identified several highly expressed genes that were not identified as highly represented among the blastemal population cells of this study (e.g. kazald1 was singled-out to be the most robust blastemal cell marker). What is the explanation for this?

Reviewer #3:

Remarks to the Author:

This research addresses the question of cellular heterogeneity in medium bud stage axolotl fore limb blastemas. The authors dissociated cells of three blastemas, performed single-cell RNA-seq on 3500 cells using the microfluidic platform inDrops, and assigned them to clusters and sub-clusters, based on similarities and differences in their gene expression patterns and gene ontology. The 10 most highly expressed genes in each cluster were identified and used to probe blastemal histological sections to spatially localize their expression. Seven transcriptionally distinct cell populations were identified in this way. Eighty percent of the cells were epidermal, comprising three distinct populations corresponding to basal epidermis, intermediate epidermis and apical (outermost) epidermis. The basal epidermis shares many molecular attributes with the intermediate epidermis and the underlying mesenchyme blastema. The intermediate layers were

the most complex, containing eight subpopulations. The four other populations were blood cells (RBCs, myeloid, lymphoid) and blastema (mesenchymal) cells.

Blastema cells express high levels of lumican (PG), follistatin-like (fstl1) and cirbp. A surprise was that only 32% of cells were positive for prrx1. The prrx1+ cells also expressed twist1, whereas the prrx1- cells did not. Also, no sox9+Twist3+ population. Overall, blastemal cells were very homogeneous in their expression profiles.

The results presented here will be of great interest to the regeneration community. The experimental protocol is convincing in demonstrating a deeper level of molecular heterogeneity in the cells of the blastema, particularly the epidermis, than previously thought. These differences presumably reflect differential functions of the several layers of epidermis that can now be analyzed more closely. The results and conclusions will be influential, especially when coupled in future with new approaches made possible by the recent publication of the axolotl genome sequence, and by the adaptation of CRISPR to edit genome expression in limb regeneration models.

I had several speculative questions as I read the paper.

(1) Does the fact that some mesenchymal cells of the blastema express cartilage-associated genes shed any light on the controversial notion that cartilage does not contribute to the blastema?

(2) I would like to have seen a more detailed description of the gene expression profile of the basal epidermis to see if any of the putative mitogens described in previous papers (AGP, Kumar et al, 2007; Kumar et al, 2011; FGF-2, Mullen et al 1996; NRG1, Farkas et al, 2016). Stocum and Dearlove (1972) should be cited as an additional reference for the in vivo interaction of the epidermis with mesenchyme in limb regeneration.

(3) Epidermal cells of skin proximal to the blastema were not compared to blastemal epidermal cells. Such data would have greatly added to the findings. What kinds of differences might there be that makes the blastemal epidermis so special?

(4) Whose dissociation method was used? There was no reference for this. I ask, because I know of only one published source.

Reviewer #4:

Remarks to the Author:

This work provided a single-cell landscape for axolotl limb regeneration. Combining single-cell RNA-seq and RNA in situ hybridization, the authors identified cell heterogeneity in the regenerating limb. This dataset covered different cell types in wound epidermis (WE) and blastema, including apical WE cells, intermediate WE cells, basal WE cells, blastema cells, red blood cells, myeloid cells and lymphoid cells. The authors highlighted the spatial expression of Krt17 during wound epidermis development. They also describe the role of myeloid cells and lymphocyte populations during blastema development. In general, this work shows interesting cell composition within both the WE and blastema. However, more works should be done before the manuscript can be accepted by Nature Communication

Major points:

1. Where is the control? The authors should do the unwounded tissues to compare with the regenerating axolotl limbs. Perhaps, a time course experiment can be provided to reveal the dynamic cell population or genetic network changes.

2. The number of cells (3500) is relatively small to resolve such heterogeneous tissues. The three

separate regenerating forelimb samples were derived from two individual axolotls. It seems that only one inDrop experiment is performed. A large group of the cells were intermediate WE cell; the number of blastema cells was not enough to uncover potential subtypes of blastema. I suggest that at least three more experiments are needed: One more for regenerating forelimb, two more for control limb or time course.

3. Fig.4 e, f. The tSNE plot and heatmap of intermediate WE cells should be carefully described. Some populations (IW2, IW3) in the heatmap are not distinct. Their actual definitions and functions should be discussed in details.

4. What are the lineage relationships for Intermediate, basal and apical WE cell (IW1, IW8)? Can you infer that through pseudo-time analysis with the single cell data?

5. Interaction between immune cells and blastema cells should be of great interest. A cellular network (receptor ligand analysis) may help to characterize the interaction among WE, immune and blastema cells.

6. In many figures, "scale bar" was mistakenly mentioned as "error bar".

7. How would tissue digestion method affect cell number ratio in the data? For example, the number of blastema cells appears to be under-represented. Is it caused of tissue digestion method?

8. When mentioning single cell RNA-seq, several important papers should be cited, for example: Macosko, et al. (2015). Cell 161, 1202–1214. & Han, et al. (2018). Cell. 172(5):1091-1107.

9. In Fig5c-5g and Fig6d-6i, the authors showed dynamic changes of cells during regeneration. Why only medium bud-stage blastema is analyzed by single cell RNA-seq?

10. "Three separate regenerating forelimb samples, derived from two individual axolotls". One axolotl was sampled twice? Are the samples from the same limb? The batch information in the study is very unclear. How would different batches affect WE populations in the data?

Minor points:

1. Heat map should have the color code legend.

2. What is the y axis of violin plot? Is it logged expression level?

3. The axolotls are described as "22-24 cm". What are the ages?

Reviewers' comments:

Reviewer #1 (Remarks to the Author):

This paper explores the cellular composition of the regenerating axolotl limb using a single cell RNAseq approach, which is complemented by in situ hybridizations with probes against a number of selected transcripts. Determining the extent of heterogeneity of the axolotl limb during regeneration is an important and timely aim. However, the manuscript is preliminary at this stage and does not live up to its rather pompous title. In contrast to the authors, I would argue the paper does not provide “a rich resource.....for axolotl limb regeneration”. As I see it, the main problem is the very low representation of mesenchymal non-blood blastemal cells in the sample, which appears to be well below 10% (I did not find the exact figure). The blastema is the origin of the regenerating limb but this low number of sequenced cells does not allow to generate a proper representation of its composition. The conclusion that the blastemal cells are “highly homogenous” is not well supported by the low number of sequenced blastemal cells. Thus, the progress in our understanding of limb regeneration provided by the paper is quite limited in that sense. The strong part of the paper is the characterization of the wound epidermis, since the vast majority of RNAseq data are derived from that tissue. While the wound epidermis data are solid, they don't provide significantly new insights into the biology of the wound epidermis. The finding that the wound epidermis can be subdivided is not new (see Tassava, 2000, which the authors also refer to) even though the authors provide additional markers for the previously defined regions. The analyses of myeloid and cells and lymphocytes is relevant but here the validation of the RNAseq data is weak. I am also surprised how the authors used GO terms. GO terms are often unspecific and the authors highlight particularly broad terms such as “negative regulation of biological processes”. A way forward could be to generate more RNAseq data, particularly of blastemal cells, or to carry out functional studies with selected candidates and use the single cell data as an entry point to these studies.

We agree that a more extensive analysis of blastema mesenchyme would provide essential information about axolotl limb regeneration. We have repeated experiments using a more prolonged enzymatic digestion and discarding portions of the wound epidermis to ensure that more mesenchymal cells were released from the blastema and captured for single cell RNA-seq. The revised manuscript describes a variety of blastema cells that are of mesenchymal origin. In addition, the inclusion of a cells collected from homeostatic limbs allows us to provide genes that are dynamically regulated in the wound epidermis during regeneration. This provides new information about how the cells of the wound epidermis may be contributing to limb regeneration. In addition, we describe molecular markers for putative ionocytes, Langerhans cells, and Leydig cells, which will allow for future study of these populations in the axolotl.

We have also eliminated the GO analysis, as we believe it is now beyond the scope of this manuscript and, as originally presented, we agree it provided limited novel insights into the populations we uncovered. Once published, the data will be available to anyone who wishes to perform additional analyses.

--

Reviewer #2 (Remarks to the Author):

This paper describes a study wherein single-cell RNA-seq was performed on 3500 cells collected from 3 medium-bud blastemal staged limbs of two adult axolotls. This is an important advance for the axolotl model and regenerative biology more generally as studies have primarily used whole tissues for gene expression analyses of regeneration. The work is original, timely and broadly interesting, and is likely to motivate similar studies in the field. My comments below are meant to improve the presentation of results from this important study.

1) The manuscript could benefit from a bit more consistency in terminology. In the introduction, the paper uses “cell types” to describe cells that were discovered by RNA-seq to exhibit the same/similar pattern of gene expression. However, in the results, the paper uses “cell populations”. Because the study only sampled a single developmental time point within the regeneration process, from only three limbs, it is probably more accurate to refer to the different cell clusters as populations. As another example, “unbiased clustering” and “unsupervised clustering” are used to describe the method for identifying cell populations.

At some point in the paper it would be helpful to briefly state what is meant by unbiased clustering. Does this apply to the algorithm used to define clusters as well as the method used to determine cluster number?

We appreciate the fact that improved terminology and description of the algorithms used would improve the clarity of the manuscript. Though we have now included a time course analysis, we have decided to follow the reviewer's advice and use cell populations to describe the cells throughout the manuscript. We have now included a sentence that describes unbiased clustering (see Introduction) and clarified that the cluster number is determined by "over-clustering" (i.e. making many populations that have very subtle transcriptomic differences). This "over-clustering" is followed by examining cluster distinctiveness using a random forest classifier that is implemented via the Seurat package, which results in merging populations that are highly similar. In general, cluster number is to an extent heuristically determined by the user while the actual process of identifying distinct clusters is not driven by any previous information about markers of specific cell populations. We have added the above explanation to the methods.

2) Several of the Gene Ontology terms that were identified for cell populations are highly inclusive of many biological processes and care should be used in their interpretation. On lines 108-110, GO: Regulation of Biological Process does not necessarily equate to regulation of growth and proliferation, and GO:Multicellular Organism Process necessarily equate to tissue and appendage morphogenesis. There are other examples in the paper.

As noted above, we have eliminated the GO analysis, as we believe it is beyond the scope of this manuscript and does not add a significant advance to the understanding of the cell populations described.

3) One of the genes discovered as a marker of intermediate WE (*pinlyp*) was recently shown to be highly differentially expressed among replicate samples during axolotl limb regeneration. Did this gene show such a pattern in this study and is there concern that some genes may exhibit bursting or on/off phenotypes among cells when performing single-cell analyses? I raise this issue because there is no general discussion of caveats (excepting Line 377 where the importance of a single cell phenotype is discussed, which isn't really worth discussing) to single-cell sequencing of a moderate number of cells from few samples. Seems like cell number, sequencing depth, dealing with a large, unannotated genome, etc, should deserve a bit more attention.

This is an interesting point and one we can answer more definitively now that we have six samples at a single time point. We evaluated average expression of *pinlyp* in the population that is defined by this gene. We found that this gene does not appear to be bursting on/off within any populations. Since single-cell RNA-seq has such a low capture efficiency it would be difficult to determine definitively if the presence of absence of a gene is due to biological or technical effects. While this is an interesting point and should be more thoroughly evaluated within this dataset, we believe at this time it is beyond the scope of this manuscript. In addition, since we are evaluating whole transcriptomes we don't expect that a few bursting genes would substantially disrupt clustering. We anticipate that researchers with interests in specific genes will use our datasets to explore those genes more thoroughly once this work is publically accessible.

We completely agree that there are caveats to single-cell RNA-seq as a technique, as we have now included a section in the discussion that provides caveats for the interpretation of our data. While we cannot perform RNA *in situ* hybridization on every gene we detect as differentially expressed, we performed these *in situ* experiments so that we could be more confident in the RNA-seq data, since this technique is still limited from the technical side in terms of capture efficiency, sequencing depth, as well as in regards to the best ways to perform the computational analysis. We anticipate that our data could be re-analyzed when future computational advances allow for more insights.

4) The paper could make a more valuable contribution to the axolotl and more generally amphibian models of regeneration if there was more integration of information about the biology of amphibian skin. There are in depth histological descriptions of amphibian skin and cell types (some of these are outside of the regeneration field but applicable, even considering differences in wound epidermis and normal epidermis). I see no reason to awkwardly introduce an interest in genes expressed during mouse development or during regeneration of gerbil skin (lines 142-145), or to focus on mammalian stem cells (Lines 292-293) when such cells are known for amphibian skin.

We have now included more discussion about amphibian skin to help put into context these data. However, due to the vast amount of data we have added, the skin has become one of many aspects of the revised paper, and we have done our best to put these data into context without over-emphasizing any one tissue type.

5) Lines 202-206 offer a “rapid evolution” explanation for not identifying gene ontologies for the myeloid population. This seems a bit of a stretch. If orthologs can be identified, then appropriate DNA sequence comparison tests could be performed. If many of these genes are unannotated/anonymous, then perhaps write that the number of unannotated/anonymous genes for this cell population is higher than for other cell populations.

This is a great point and one we have re-evaluated. The number of unannotated genes is not particularly high in the immune cell populations, thus it is likely that axolotls have different markers for immune cells than more highly studied species. Since most immunologic work has been performed in mammals (e.g. mice and human) it will be important to follow up more thoroughly on how to properly define axolotl immune cells. Thus, we have removed the “rapid evolution” explanation and replaced it with a line stating that different species such as axolotl may have fundamentally different marker genes for immune cells.

6) In the section describing T cells, it would be good to cite the recent paper by Hui et al in Developmental Cell showing the requirement of Tregs in multiple examples of zebrafish organ regeneration.

This is a new and exciting paper that we have now added to the paper. In general, we highlighted changes suggested by the reviewers, but thought that highlighting all changes would be unhelpful, as almost the entire results section and large portions of the discussion have been overhauled.

7) Line 282: Probably the word “diversity” is better than “contamination”.

We agree and have removed this line.

8) Line 320: Macrophages have an early role in regeneration but their activities do not initiate regeneration.

We have changed this line to read that macrophages are required for limb regeneration and not for the initiation of regeneration per se.

9) Line 351 suggests that the single cell data presented in the paper provides a new lens to re-evaluate bulk tissue experiments. It would improve the paper if this were addressed explicitly with examples that concern estimates of gene expression and not issues concerning gene orthology, which is a separate issue. For example, a recent transcriptome analysis of the axolotl blastema (Bryant et al in Cell Reports) identified several highly expressed genes that were not identified as highly represented among the blastemal population cells of this study (e.g. *kazald1* was singled-out to be the most robust blastemal cell marker). What is the explanation for this?

We have evaluated the expression of the *kazald1* called out by Bryant et al in our dataset. While it was blastema specific, it was not highly expressed. We suspect that this may be due to the 3' capture method employed by inDrops, which at a maximum may capture ~10% of transcripts. We have noted this in the manuscript and expect that future technological advances will likely shed light on this subject.

--

Reviewer #3 (Remarks to the Author):

This research addresses the question of cellular heterogeneity in medium bud stage axolotl fore limb blastemas. The authors dissociated cells of three blastemas, performed single-cell RNA-seq on 3500 cells using the microfluidic platform inDrops, and assigned them to clusters and sub-clusters, based on similarities and differences in their gene expression patterns and gene ontology. The 10 most highly expressed genes in each cluster were identified and used to probe blastemal histological sections to spatially localize their expression. Seven transcriptionally distinct cell populations were identified in this way. Eighty percent of the cells were epidermal, comprising three distinct populations corresponding to basal epidermis, intermediate epidermis and apical (outermost) epidermis. The basal epidermis shares many molecular attributes with the intermediate epidermis and the underlying mesenchyme blastema. The intermediate layers were the most complex, containing eight

subpopulations. The four other populations were blood cells (RBCs, myeloid, lymphoid) and blastema (mesenchymal) cells.

Blastema cells express high levels of lumican (PG), follistatin-like (fstl1) and cirbp. A surprise was that only 32% of cells were positive for prrx1. The prrx1+ cells also expressed twist1, whereas the prrx1- cells did not. Also, no sox9+Twist3+ population. Overall, blastemal cells were very homogeneous in their expression profiles.

The results presented here will be of great interest to the regeneration community. The experimental protocol is convincing in demonstrating a deeper level of molecular heterogeneity in the cells of the blastema, particularly the epidermis, than previously thought. These differences presumably reflect differential functions of the several layers of epidermis that can now be analyzed more closely. The results and conclusions will be influential, especially when coupled in future with new approaches made possible by the recent publication of the axolotl genome sequence, and by the adaptation of CRISPR to edit genome expression in limb regeneration models.

I had several speculative questions as I read the paper.

(1) Does the fact that some mesenchymal cells of the blastema express cartilage-associated genes shed any light on the controversial notion that cartilage does not contribute to the blastema?

This is an interesting point and one we were also curious about. Now that we have dramatically increased the number of blastema progenitor cells in the study, we believe we can make a more firm speculation about cartilage-derived blastema cells. While cells within the blastema progenitor populations expressed cartilage-associated markers they also expressed markers of the fibroblast-lineage. Thus, we expect that these are likely fibroblast-derived cells, and not cells that are derived from the cartilage, which then acquire fibroblast-specific markers. In addition, the use of large adult animals that have completely ossified long bones likely makes the presence of cartilage at the amputation plane unlikely. Thus, future studies using lineage tracing or amputation through tissue that contains cartilage may shed more light on this question. That being said, as noted by the reviewer, this is in line with the current literature.

(2) I would like to have seen a more detailed description of the gene expression profile of the basal epidermis to see if any of the putative mitogens described in previous papers (AGP, Kumar et al, 2007; Kumar et al, 2011; FGF-2, Mullen et al 1996; NRG1, Farkas et al, 2016). Stocum and Dearlove (1972) should be cited as an additional reference for the in vivo interaction of the epidermis with mesenchyme in limb regeneration.

We are also very interested about the potential contributions of the basal WE as suggested by the reviewer and have made a point to highlight this epidermal population. We have specifically looked at these markers to help put into context the gene expression of other studies in regards to the cell populations we describe. We have also added this reference, and thank the reviewer for pointing it out.

(3) Epidermal cells of skin proximal to the blastemal were not compared to blastemal epidermal cells. Such data would have greatly added to the findings. What kinds of differences might there be that makes the blastemal epidermis so special?

This is a very interesting question, and one we hope can be answered by our addition of homeostatic limb samples. We have called out regeneration-induced markers in the basal WE. We hope that this sheds some light on the uniqueness of the epidermis overlaying the blastema. We also plan on making a website that allows for query of gene expression profiles over time, which should allow for query of gene expression that changes over the course of regeneration, in all populations, including the epidermal populations.

(4) Whose dissociation method was used? There was no reference for this. I ask, because I know of only one published source.

We have added references for the dissociation method. We modified protocols from Goetz et al 2012, Kragl et al 2013 Dev Bio, and Briggs et al 2018. In addition, we have written a detailed protocol and posted it on the protocol repository protocols.io.

--

Reviewer #4 (Remarks to the Author):

This work provided a single-cell resolution landscape for axolotl limb regeneration. Combining single-cell RNA-seq and RNA in situ hybridization, they identify cell heterogeneity in the regenerating limb. This dataset covered different cell types in wound epidermis (WE) and blastema, including apical WE cells, intermediate WE cells, basal WE cells, blastema cells, red blood cells, myeloid cells and lymphoid cells. The authors highlighted the spatial expression of Krt17 during wound epidermis development. They also describe the role of myeloid cells and lymphocyte populations during blastema development. In general, this work shows interesting cell composition within both the WE and blastema. However, more work should be done before the manuscript can be accepted by Nature Communication

Major points:

1. Where is the control? The authors should do the unwounded tissues to compare with the regenerating axolotl limbs. Perhaps, a time course experiment can be provided to reveal the dynamic cell population or genetic network change.

This is an important undertaking, and we have now included a time course that includes limb cells at homeostasis, and post-amputation during wound healing, blastema development, and the blastema prior to re-differentiation.

2. The number of cells (3500) is relatively small to solve such heterogeneous tissues. The three separate regenerating forelimb samples were derived from two individual axolotls. It seems that only one inDrop experiment is performed. A large group of the cells were intermediate WE cell; the number of blastema cells was not enough to uncover potential subtypes of blastema cells. I suggest that at least three more experiments are needed: One more for regenerating forelimb, two more for control limb.

We have now included almost 10 times as many cells as our previous submission, which we hope can provide more insight into blastema subpopulations.

3. Fig.4 e, f. The tSNE plot and heatmap of intermediate WE cells should be carefully described. Some populations (IW2, IW3) in heatmap are not distinct. Their actual definition and function should be discussed in details.

This is a great point and we think that due to the outward differentiating nature of the epidermis that the intermediate WE contains a variety of cells in transition and not necessarily highly distinct sub-populations. As noted below, we have now performed pseudotime analysis on the WE, which we think provides a better understanding of both lineage relationships and the true differential cell populations present in the WE.

4. What are the lineage relationships for Intermediate, basal and apical WE cell (IW1, IW8)? Can you infer that through pseudo-time analysis with the single cell data?

Great suggestion, this has now been included.

5. Interaction between immune cells and blastema cells should be of great interest. A cellular network (receptor ligand analysis) may help to characterize the interaction among WE, immune and blastema cells.

We agree that this is of utmost importance. However, we are hesitant to make conclusions on potential interactions between different cell populations without more spatial information. We believe that the emerging *in situ* RNA-seq technologies (e.g. MERFISH and others) may be able to better address this important question.

6. In many figures, “scale bar” was mistakenly mentioned as “error bar”.

We regret this error and have now made the appropriate changes.

7. How would tissue digestion method affect cell number ratio in the data? For example, the number of blastema cells appears to be under represented, is it a result of tissue digestion method?

In our initial submission we were very concerned about the speed of digestion and processing of the tissue. We believe this led us to capture an overabundance of cells that were easy to digest (i.e. WE and immune cells). We have modified this protocol for longer digestion and mechanical removal of half or all of the WE to ensure that blastema cells were captured. Due to the differences in tissue architecture this

appears to have helped capture more blastema, but making conclusions about cell quantity between the WE and blastema is difficult since the WE was removed.

8. When mentioning single cell RNA-seq, several important papers should be cited, for example: Macosko, et al. (2015). Cell 161, 1202–1214. & Han, et al. (2018). Cell. 172(5):1091-1107.

These seminal papers have been added.

9. In Fig5c-5g and Fig6d-6i, you showed us the dynamic changes of cells during regeneration. Why only medium bud-stage blastema is analyzed by single cell RNA-seq?

As noted above, we have now included a time course of homeostatic and regenerating time points.

10. “Three separate regenerating forelimb samples, derived from two individual axolotls”. One axolotl was sampled twice? Are the samples from the same limb? The batch information in the study is very unclear. How would different batches affect WE populations in the data?

This has been clarified in the methods. In general, we have now included 13 unique biological samples all from individual animals.

Minor points:

1. Heat map should have the color code legend.

We regret this error, and have removed heatmaps from this revision.

2. What is the y axis of violin plot? Is it logged expression level?

This is \log_2 values of normalized expression and we have added this to the figure legends.

3. The axolotls are described as “22-24 cm”. What is the age?

We have added age information for the axolotls. We also added important information about the fact that these animals had never been previously amputated and had been individually housed prior any cannibalistic biting of each other's limbs.

Reviewers' Comments:

Reviewer #1:

Remarks to the Author:

The revised version of this paper has significantly improved compared to the first submission. Although still no functional data are presented, which is clear minus, the authors have performed additional sequencing and are presenting their data in a more coherent way. The pseudotime analysis is also nice. Although it is relevant to propose putative developmental trajectory within the blastema and wound epidermis but I disagree that data supports claims regarding the origin of the different blastemal cells. Concluding cell origin from the stump would require combination of cell tracing with the single cell RNAseq, and no such experiments have been performed. Thus, any claim on cell origin should be systematically removed or appropriately reworded both in main text and figure legends.

The wording of the paper is occasionally ambiguous and would need further editing for clarity. A couple of examples:

- The headline "Cellular sources of nerves and blood vessels" is unclear. No new nerves are formed in the blastema. I guess the authors mean nerve associated cells such as Schwann cells.
- Line 305 "To begin to understand potential the regenerative trajectories in the blastema, we selected cells from all regenerating time points that we considered to reside within the blastema. This included all FDBs, Schwann cells, endothelial cells, myogenic blastema cells, and pericytes." What do the authors mean here? They selected all cell types that they identified cumulatively at the different time points of regeneration? But earlier in the manuscript the authors write that there are cells which don't fall under either of the above listed cell types: (line 301) "...fall under the generic label of blastema cell due to the fact that they reside within the structure that is the blastema." Why were those cells not included here?
- Line 234: "Many of these cells lack canonical markers for immune cells that are used in other species." I guess the authors mean markers that are used to identify immune cells in other species.

Reviewer #2:

Remarks to the Author:

The authors satisfactorily addressed concerns I raised in the original submission, and the revised submission includes more data and analyses, which further strengthens the study. The work is original, timely and broadly interesting, and is likely to motivate similar studies in the field.

Reviewer #4:

Remarks to the Author:

Great job ! This should be a cover article.

REVIEWERS' COMMENTS:

Reviewer #1 (Remarks to the Author):

The revised version of this paper has significantly improved compared to the first submission. Although still no functional data are presented, which is clear minus, the authors have performed additional sequencing and are presenting their data in a more coherent way. The pseudotime analysis is also nice. Although it is relevant to propose putative developmental trajectory within the blastema and wound epidermis but I disagree that data supports claims regarding the origin of the different blastemal cells. Concluding cell origin from the stump would require combination of cell tracing with the single cell RNAseq, and no such experiments have been performed. Thus, any claim on cell origin should be systematically removed or appropriately reworded both in main text and figure legends.

We agree that making definitive claims about the cellular origins within the stump cannot be derived from this dataset and attempted to try to not make these claims absolute. We have modified the text further to make it clear that more work would be required to elucidate the origins within the stump. In line with this, we have changed fibroblast-derived blastema to fibroblast-like blastema throughout the text to ensure there is no confusion about our interpretation.

The wording of the paper is occasionally ambiguous and would need further editing for clarity. A couple of examples:

- The headline “Cellular sources of nerves and blood vessels” is unclear. No new nerves are formed in the blastema. I guess the authors mean nerve associated cells such as Schwann cells.

Thanks for noting this error. Yes, we do mean nerve-associated cells and have changed this section header.

- Line 305 “To begin to understand potential the regenerative trajectories in the blastema, we selected cells from all regenerating time points that we considered to reside within the blastema. This included all FDBs, Schwann cells, endothelial cells, myogenic blastema cells, and pericytes.” What do the authors mean here? They selected all cell types that they identified cumulatively at the different time points of regeneration? But earlier in the manuscript the authors write that there are cells which don’t fall under either of the above listed cell types: (line 301) “...fall under the generic label of blastema cell due to the fact that they reside within the structure that is the blastema.” Why were those cells not included here?

We have clarified this section to note that we took all non-immune cells that reside within the blastema. The only exception being the exclusion of tenocytes which only had 11 cells and would be difficult to make conclusions on such a small number of cells.

- Line 234: “Many of these cells lack canonical markers for immune cells that are used in other species.” I guess the authors mean markers that are used to identify immune cells in other species.

We have clarified this sentence and thank the reviewer for pointing out these ambiguous sections. In general, we have corrected these areas as well as given the manuscript a final, thorough edit to remove other occurrences of ambiguity.

--

Reviewer #2 (Remarks to the Author):

The authors satisfactorily addressed concerns I raised in the original submission, and the revised submission includes more data and analyses, which further strengthens the study. The work is original, timely and broadly interesting, and is likely to motivate similar studies in the field.

Thank you for your insightful comments and suggestions.

--

Reviewer #4 (Remarks to the Author):

Great job ! This should be a cover article.

Thanks, we appreciate the thorough and helpful review.